# Antitumoral Activities of Curcumin and Recent Advances to ImProve Its Oral Bioavailability

**DOI:** 10.3390/biomedicines9101476

**Published:** 2021-10-14

**Authors:** Marta Claudia Nocito, Arianna De Luca, Francesca Prestia, Paola Avena, Davide La Padula, Lucia Zavaglia, Rosa Sirianni, Ivan Casaburi, Francesco Puoci, Adele Chimento, Vincenzo Pezzi

**Affiliations:** Department of Pharmacy and Health and Nutritional Sciences, University of Calabria, Via Pietro Bucci, Arcavacata di Rende, 87036 Cosenza, Italy; nocitomarta90@tiscali.it (M.C.N.); ariannadl@hotmail.it (A.D.L.); francescaprestia908@gmail.com (F.P.); paola.avena@unical.it (P.A.); davidelapadula@live.it (D.L.P.); luciazavaglia@hotmail.it (L.Z.); rosa.sirianni@unical.it (R.S.); ivan.casaburi@unical.it (I.C.); francesco.puoci@unical.it (F.P.)

**Keywords:** curcumin, cancer cells, bioavailability, curcumin derivatives, curcumin analogues, curcumin delivery systems

## Abstract

Curcumin, a main bioactive component of the *Curcuma longa* L. rhizome, is a phenolic compound that exerts a wide range of beneficial effects, acting as an antimicrobial, antioxidant, anti-inflammatory and anticancer agent. This review summarizes recent data on curcumin’s ability to interfere with the multiple cell signaling pathways involved in cell cycle regulation, apoptosis and the migration of several cancer cell types. However, although curcumin displays anticancer potential, its clinical application is limited by its low absorption, rapid metabolism and poor bioavailability. To overcome these limitations, several curcumin-based derivatives/analogues and different drug delivery approaches have been developed. Here, we also report the anticancer mechanisms and pharmacokinetic characteristics of some derivatives/analogues and the delivery systems used. These strategies, although encouraging, require additional in vivo studies to support curcumin clinical applications.

## 1. Introduction

Curcuma is one of the largest genera in the Zingiberaceae family which comprises approximately 133 species [1]. It is widely distributed in the tropical regions spanning India to Southern China and Northern Australia [1]. The species of greatest interest is *Curcuma longa* L., which is cultivated particularly in India. It consists of an underground root (rhizome) from which, once dried and ground, a powder with a characteristic yellow-orange color is obtained. Curcumin extract is composed of three curcuminoids at different proportions such as curcumin (1, 7-bis (hydroxyl-3-methoxyphenyl)-1,6- heptadiene-3, 5-dione) (curcumin I) (~ 77%), demethoxy curcumin (curcumin II) (17–18%) and bis-demethoxycurcumin (curcumin III) (3–5%) (Figure 1) [2]. The volatile fraction is quantitatively important, which mainly contains several terpenic compounds including α-zingiberene, curlone and α-turmerone [3]. Among the curcuminoids, the polyphenol curcumin is the most active. It possesses a wide range of pharmacological proprieties [4,5], acting as an antimicrobial [6], antiviral [7], antifungal [8], antioxidant [9], antimalarial [10], anti-inflammatory [11], antiaging [12] and antitumoral [13] agent. In recent years, extensive research indicated that this polyphenol can prevent and suppress tumor initiation, promotion and progression and can be used to treat cancer by interfering with several signaling pathways [13,14]. At a molecular level, curcumin’s anticarcinogenic effects are attributed to its ability to modulate transcription factors, growth regulators, adhesion molecules, apoptotic genes, angiogenesis regulators, and more [14]. In vitro studies demonstrated that curcumin suppresses the proliferation of a wide variety of tumor cells; downregulates the expression of transcription factors (NF-kB, AP-1 and Egr-1), COX2, MMPs, TNF, chemokines, cell surface adhesion molecules and cyclins, growth factor receptors (such as EGFR and HER2); and inhibits tyrosine and serine/threonine kinases (i.e., JNK) activity [14]. By inhibiting STAT3, NF-kB and WNT/β-catenin signaling pathways, curcumin interferes with cancer development and progression [15,16,17]. In addition, the inhibition of Sp-1 and its regulated genes may serve as an important mechanism to prevent cancer formation, migration, and invasion [14]. Additionally, curcumin induces apoptosis through both mitochondria-dependent [18] as well as mitochondria-independent mechanisms [19], depending on the cell type.

Despite curcumin’s anticancer potential, its therapeutic application is limited by its poor bioavailability, due to the low or very low absorption after oral intake. After oral administration, curcumin is metabolized into curcumin glucuronide and curcumin sulfonate by the liver during phase II reactions. Glucuronidation and sulfation increase curcumin’s water solubility, but decrease its effectiveness and accelerate its removal via urine [20]. Specifically, the different factors contributing to the low bioavailability include the low plasma level, tissue distribution, poor absorption, high rate of metabolism, inactivity of metabolic products, rapid elimination and clearance from the body [21]. In order to improve curcumin pharmacokinetic characteristics and then biological activity, synthetic derivatives/analogues [22,23,24] and various drug delivery strategies have been developed [25,26,27,28,29,30].

In this review, we summarize the recent advances in molecular mechanisms activated by curcumin and the elicited inhibition of cancer cell growth and progression. Additionally, the improved pharmacokinetic proprieties of curcumin derivatives/analogues, and the attempted targeted and triggered drug delivery systems, are reviewed.

## 2. Antitumoral Activities of Curcumin

Dietary polyphenolic phytochemicals are able to block tumor initiation or progression [31]. Several studies indicated that curcumin can effectively prevent and treat cancer at the initiation, promotion and progression level [32]. This propriety is due to its ability to target critical processes primarily involved in cancer development, such as proliferation, apoptosis and metastasis.

### 2.1. Antiproliferative Effects of Curcumin

Some molecular alterations associated with carcinogenesis occur in signaling pathways regulating cell proliferation [31]. Accumulating data show that curcumin displays in vitro antiproliferative effects on several types of cancer such as breast, colorectal, bladder, brain and gastrointestinal through several mechanisms of action [33] (see Table 1 for a summary of the data). Generally, the antiproliferative effects of curcumin could be attributed to its ability to regulate cell cycle progression, protein kinases activity and transcription factor expression.

In breast cancer cell lines, curcumin inhibits proliferation by reducing cyclin D1 expression at mRNA and protein levels, consequently decreasing CDK4 activity [34,35]. In MCF-7 breast cancer cells, curcumin caused G1 cell cycle arrest, cyclin E proteosomal degradation, p53 upregulation and an increase in p21 and p27 CDK inhibitor levels [36]. In MCF-7 and MDA-MB-231 breast cancer cells, a G2/M cell cycle arrest by curcumin was also reported. Specifically, following GSK3β upregulation, a loss of nuclear β-catenin produced a downregulation of its downstream target, cyclin D1 [37]. Additionally, curcumin is a potent inhibitor of NF-kB [16], a critical regulator of cell proliferation and survival [38]. Liu et al. demonstrated that in MCF-7 cells, curcumin markedly decreased cell proliferation in a concentration- and time-dependent manner by regulating the NF-kB signaling, as shown by a decrease in p65 and an increase in IκB protein expression [16]. Moreover, Zhou et al. demonstrated that, in MCF-7 cells and MCF-7 xenografts, a combination of curcumin with the chemotherapeutic agent mitomycin C produced cell cycle arrest at the G1 phase as a consequence of decreased cyclin D1, cyclin E, cyclin A, CDK2, and CDK4 expression, along with the induction of the cell cycle inhibitor p21 and p27 [35]. The potent inhibitory effect produced by curcumin and mitomycin C combination on MCF-7 growth was also observed in MCF-7- and MDA-MB-231-derived cancer stem cells [39]. In addition, curcumin was proven to be effective to sensitize MDA-MB-231 and MCF-7 cells to commonly used chemotherapeutic agents paclitaxel, cisplatin and doxorubicin [39]. In a recent work, it has been shown that curcumin inhibited the growth of MDA-MB-231 and MDA-MB-468 triple-negative breast cancer cells by reducing the expression of histone methyl transferase EZH2 and reactivating that of DLC1 [40]. EZH2 is an oncogenic factor commonly upregulated in human cancers [41], while DLC1 is a downregulated tumor suppressor in many malignant tumors [42].

In vitro studies showed that curcumin significantly inhibited colon cancer cell growth modulating multiple molecular targets and distinct signaling pathways [43,44]. A study performed by Lim et al. demonstrated that in a HCT-116 human colon cancer cell line, curcumin downregulated CDK2 expression, causing G1 phase arrest [45]. Moreover, in the same cell line, curcumin enhanced ROS generation and downregulated the transcription factor E2F4 and its target genes, including cyclin A, p21 and p27 [46]. Another study by Watson et al. investigated curcumin cytotoxicity in HCT-116 and HT29 cell lines, showing a time- and dose-dependent cell proliferation inhibition when p53 was upregulated [47]. Curcumin prevented cell proliferation by cell cycle arrest at the G2/M phase in both HCT116 p53+/+ and p53−/− cells in a concentration-dependent manner and induced senescence accompanied by autophagy [48]. Recently, Calibasi-Kocal et al. demonstrated antiproliferative, wound healing, anti-invasive and antimigratory effects of curcumin not only on HCT-116 cells, but also on the LoVo metastatic colorectal cancer cell line, confirming its anticancer activity [44].

The antitumoral effects of curcumin were also demonstrated in bladder cancer cells. Curcumin’s inhibitory effects were observed in T24 [49,50], 5637 [50] and in RT4 [49] human bladder cancer cell lines. In T24 and RT4 cells, a decrease in Trop2, a cell surface receptor that transduces via calcium signals, is required for curcumin-mediated effects including inhibition of cell proliferation and migration, apoptosis and cell cycle arrest. A Trop2 decrease caused a reduction in the expression levels of its downstream target cyclin E1, and an increase in p27 levels [49].

Curcumin has been shown to decrease malignant characteristics of glioblastoma cells acting on several targets, including growth factor receptors, kinases, transcriptional factors and inflammatory cytokines [51]. In the U87 human glioma cell line, curcumin inhibited proliferation-suppressing cyclin D1 and induced p21 expression. In these cells, curcumin via ERK and JNK signaling induced expression of the transcription factor Egr-1 which, in turn, increased p21 gene expression [52]. In the same cells, curcumin induced G2/M cell cycle arrest and apoptosis by increasing FoxO1 expression [53]. In U251 and SNB19 human glioblastoma cells curcumin reduced the expression of Skp2 and upregulated p57 [54]. Skp2, a component of the ubiquitin proteasome system, is responsible for ubiquitin-mediated degradation of the cell cycle inhibitors p21, p27 and p57 [55]. Recently, Luo et al. demonstrated that curcumin decreased proliferation of U251 and LN229 human glioblastoma cells via inhibition of HDGF [56], an angiogenesis-promoting growth factor commonly upregulated in gliomas. In human glioma cell lines U251 and LN229, HDGF forms a complex with β-catenin promoting tumor growth and metastasis. Curcumin significantly reduced HDGF expression, and consequently its binding to β-catenin [56]. Moreover, several studies provided evidence that curcumin potentiated the effects of chemotherapy and radiation therapy while protecting normal tissue, selectively inducing cell death in glioblastoma cancer cells [57,58].

In gastrointestinal cancer cell lines, curcumin inhibited ODC and increased SMOX activity, two key enzymes in polyamine synthesis and catabolism, respectively. Importantly, elevated levels of polyamines have been associated with several cancers, and altered levels of the rate limiting enzymes in both biosynthesis and catabolism have been observed. The combination of curcumin with ODC inhibitor DFMO significantly potentiated ODC inhibition decreasing AGS gastric adenocarcinoma cell growth [59]. Evidence regarding curcumin’s potential in gastric cancer prevention has been accumulating. Curcumin induced cell-cycle arrest at the G1 phase by downregulating cyclin D1 expression in BGC823, SGC7901, MKN1 and MGC803 gastric cancer cells by inhibiting EGF/PAK1/NF-kB pathway [60]. Moreover, curcumin exerted antiproliferative effects by inhibiting the Wnt/β-catenin signaling pathway. In SNU1, SNU-5 and AGS human gastric cancer cells, curcumin caused a reduction in Wnt3a, LRP6, phospho-LRP6, β-catenin, phospho-β-catenin, c-Myc and survivin [61]. The antiproliferative effects of curcumin were also demonstrated in SGC-7901 and BGC-823 gastric cancer cells by activating p53/p21 and inhibiting PI3K signaling pathways [62]. In another work, SGC7901 gastric cancer cell proliferation was reduced after curcumin treatment via c-Myc/long non-coding RNA (lncRNA) H19 downregulation and p53 upregulation [63]. Studies reported that H19, an oncogenic lncRNA [64], is abnormally upregulated in gastric cancer and contributes to cellular proliferation by directly inactivating p53 [65]. Furthermore, the oncogene c-Myc was shown to directly induce H19 expression by binding to the H19 promoter, and thereby promoting proliferation of gastric cancer cells [66]. A recent investigation defined curcumin effects on microRNAs expression related to gastric cancer cell proliferation [67]. In SGC-7901 cells, curcumin inhibited cell cycle progression and induced apoptosis by upregulating miR-34a, which decreased CDK4 and cyclin D1 protein expression. The same effects were obtained by transfecting the cells with specific miR-34a agomir [67]. In BGC-823 and SGC7901 gastric cancer cells, curcumin inhibited cell proliferation and increased cell death by upregulating miR-33b which, in turn, decreased expression of the apoptosis inhibitor XIAP by targeting its 3′ UTR [68].

Mounting evidence indicates that curcumin affects several molecular pathways involved in melanoma pathogenesis, making it a promising therapeutic agent to be used against this type of cancer [69]. This phytochemical compound was able to arrest cell cycle at the G2/M phase by inhibiting NF-kB and iNOS activity in human melanoma A375 and MeWo cells [70]. Another study postulated that curcumin’s effect on cell cycle arrest at the G2/M phase was dependent on PDE inhibition, an enzyme that catalyzed cAMP and/or cGMP hydrolysis. Specifically, curcumin decreased cell proliferation and cell cycle progression by inhibiting PDE1A, cyclin A, UHRF1 and DNMT1 expression while increasing that of p21 and p27 in B16F10 murine melanoma cells [71]. Another report showed that curcumin caused cell cycle arrest at the G2/M phase, and induced autophagy by downregulating Akt/mTOR axis in human melanoma A375 and C8161 cell lines [72]. Curcumin antiproliferative effects were also observed in other three melanoma cell lines (C32, G-361, and WM 266-4), all of which are characterized by B-Raf mutations. In these cells, curcumin antitumor effects were associated with the suppression of NF-kB and IKK activity but were independent from the inhibition of B-Raf/MEK/ERK and Akt pathways [73].

### 2.2. Pro-Apoptotic Effects of Curcumin

In addition to antiproliferative properties, curcumin shows extensive therapeutic potential as an apoptotic inducer through several mechanisms demonstrated in different cancer cell models (see Table 1 for a summary of the data). In MCF-7 breast cancer cells, a sub-cytotoxic dose of curcumin induced apoptotic cell death through an increase in histone H3 acetylation and glutathionylation which, in turn, promoted the transcriptional activation of different proapoptotic genes [74]. Several reports indicated that, in breast cancer cells, curcumin induced apoptosis via p53-dependent and -independent pathways [75,76]. Patel et al. demonstrated that, in MCF-7 cells, curcumin enhanced p53 expression and activated Parp-1-mediated apoptotic pathways [77]. The p53-independent effects of curcumin were observed in cancer cells lacking a functional p53 protein such as MDA-MB-231. In these cells, curcumin induced ROS generation which altered mitochondrial membrane permeability, reduced intracellular GSH levels, increased Bax/Bcl-2 ratio and cleaved-caspase 3 expression [78]. In MDA-MB-231 cells, curcumin inhibited NF-kB p65, triggering apoptosis [79]. Moreover, in the same cell line, curcumin was able to mediate apoptotic cell death through FAS inhibition [80].

The apoptotic effects of curcumin were evident in melanoma. In A375 cells, curcumin promoted tumor cell apoptosis by inhibiting the JAK-2/STAT-3 signaling pathway and downregulating Bcl-2 [81]. Moreover, the ability of curcumin to induce apoptosis was demonstrated in four human melanoma cell lines (PMWK, Sk-mel-2, Sk-mel-28, and Mewo) with mutant p53 through the activation of FAS/caspase 8 pathway [82]. Curcumin-dependent apoptosis was also observed in HEY ovarian cancer cells where p53 knockdown or p53 inhibition did not prevent curcumin’s inhibitory effect. In these cells, apoptosis occurred through the activation of p38 MAPK, the inhibition of pro-survival Akt signaling, along with a decreased expression of Bcl-2 and survivin [83]. Additionally, in the multiple myeloma RPMI 8226 cell line, curcumin upregulated p53 and Bax protein levels and downregulated MDM2, a known p53 inhibitor [84].

In colorectal cancer cells, curcumin triggered the apoptotic process via the subsequent modulation of various target molecules [43]. In HT-29 colon cancer cells, curcumin-induced apoptosis was related to a decreased COX2 expression and AKT phosphorylation along with an increased activation of AMPK signaling [85]. Moreover, this polyphenol promoted apoptosis in HCT116, HT29, and SW620 colorectal cancer cell lines by suppressing constitutive and inducible NF-kB activity and NF-kB-regulated gene products such as Bcl-2, Bcl-xL, IAP-2, COX2 and cyclin D1 [86]. Additionally, Narayan et al. showed that curcumin inhibited the Wnt/β-catenin pathway by inducing the caspase 3-mediated cleavage of beta-catenin, E-cadherin and APC, leading to loss of cell–cell adhesion and apoptosis in HCT-116 colon cancer cells [87]. It has been reported that curcumin can activate extrinsic apoptotic pathway, upregulating DR5 protein in HCT-116 and HT-29 colon cancer cells [88]. Furthermore, curcumin triggered Fas-mediated caspase 8 activation in HT-29 cells [89]. In these cells, treatment with curcumin caused a mithocondrial [Ca^2+^] increase, cytochrome c release from mitochondria to cytosol, mithocondrial membrane potential reduction, Bax increase and Bcl-2 as well as caspase 3 and 7 activation [89]. Curcumin-induced Bcl-2 downregulation and Bax upregulation in both HCT-116 [90] and COLO-205 colon cancer cells [91].

Caspase 3/7 activity was investigated by Shi et al. in bladder cancer cells. The authors demonstrated the ability of curcumin to induce apoptosis through a caspase-dependent mechanism in two human urinary bladder carcinoma cells [50]. The same apoptotic mechanism was also observed in other cancer types such as glioblastoma [92]. A similar feature also occurs in human osteosarcoma (HOS) cells, where curcumin caused cell cycle arrest determining apoptosis, as demonstrated by caspase 3 and PARP-1 cleavage [93]. Recently, it was observed that curcumin inhibited ODC activity and polyamine biosynthesis in AGS gastric adenocarcinoma cells. In this type of cell, curcumin caused an increase in ROS levels responsible for DNA damage and, thus, apoptosis [59].

The pro-apoptotic effects of curcumin are also mediated by the modulation of miRNAs in several cancer cells. Curcumin reduced Bcl-2 expression by upregulating miR-15a and miR-16 in MCF-7 cells [94] and promoted apoptosis through the miR-186 signaling pathway in human lung adenocarcinoma cells [95]. Recently, in RT4 schwannoma cells, Sohn et al. demonstrated that curcumin enhanced the expression of apoptotic protein Bax and decreased Bcl-2, as well as determined caspase 3/9 activation. All of these events were related to curcumin-mediated miRNA 344a-3p upregulation [96].

### 2.3. Antimetastatic Effects of Curcumin

In addition to the antiproliferative and apoptotic effects, curcumin acts as an antimetastatic agent (see Table 1 for a summary of the data) [15,44]. The metastatic cascade starts with the loss of cell-to-cell and cell-to-substrate adhesion, a feature of the EMT process, which allows the acquisition of a mobile phenotype, the dissociation of cells from primary tumor and the spread to distant tissues and organs [97].

The effects of curcumin on genes involved in EMT was evaluated in breast cancer cells. It was demonstrated that curcumin decreased the gene transcription and protein expression of Axl, Slug, CD24 and RhoA, which regulate EMT and, consequently, migration and invasion of MCF-10F and MDA-MB-231 breast cancer cells. Curcumin elicited these effects through the upregulation of miR-34a, which acts as a tumor suppressor gene in both cell lines [98]. The antimetastatic effects of curcumin occur through the modulation of several signaling pathways, including NF-kB. NF-kB/p65 transcriptionally regulates TWIST1, SLUG and SIP1 which, in turn, repress E-cadherin while activating the mesenchymal markers N-cadherin and MMP11, resulting in EMT progression [99]. In MCF-7 cells, curcumin was able to inhibit uPA production by preventing NF-kB activation [100]. Through the same mechanism, curcumin downregulated CXCL1 and 2, two inflammatory cytokines involved in MDA-MB-231 breast cancer cells migration [101]. A critical event in tumor cell invasion and metastasis is the degradation of the extracellular matrix by MMPs, enzymes that degrade a range of extracellular matrix proteins, allowing cancer cells to migrate and invade [102]. Curcumin was able to inhibit LPA-activated invasion by attenuating the RhoA/ROCK/MMPs pathway in MCF-7 cells [103]. Similarly, in SO-Rb50 and Y79 human retinoblastoma cell lines, curcumin reduced migration and invasion by decreasing MMP2, RhoA, ROCK1 and vimentin expression. The authors provided evidence that, in these cells, curcumin’s antitumor activity requires miR-99a upregulation and JAK/STAT3 pathway inhibition [104]. MMP’s decrease after curcumin treatment was also observed in T24 and 5637 human bladder cancer cell lines [50]. Additionally, in T24 and RT4 bladder cancer cells, curcumin’s antimetastatic mechanism included a Trop2 decrease [49], a gene also involved in tumor aggressiveness and metastasis formation [105]. In prostate cancer cells, curcumin treatment suppressed EGF, heregulin-stimulated PC-3 and androgen-induced LNCaP cell invasion. Particularly, curcumin significantly reduced MMP9 activity and downregulated cellular matriptase, a membrane-anchored serine protease involved in tumor formation and invasion [106]. Recently, in an HCT-116 human colorectal carcinoma cell line, it has been demonstrated that the expression of proteins related to cell migration, including MMP9 and claudin-3, was downregulated by increasing doses of curcumin [107]. These data were confirmed by an independent group using the same cell line in addition to LoVo human metastatic colon cancer cells, establishing curcumin anti-invasive and antimigratory properties [44]. Furthermore, in human melanoma A375 cells, curcumin decreased MMP2 and MMP9 expression while increasing TIMP-2, a tissue inhibitor of metalloproteinases [81]. Activation of melanoma cell migration and invasion by OPN was also counteracted by curcumin. Specifically, curcumin was able to downregulate pro-MMP2 activation by preventing OPN-mediated NF-kB nuclear translocation [108]. The ability of curcumin to interfere with NFk-B pathway was also evident in Hela cervical cancer cells, where it was also demonstrated an effect on Wnt/β-catenin signaling, two pathways involved in proliferation and invasion of cervical cancer [17]. It has been shown that STAT3 activation is associated with metastasis formation in several tumors [109]. In SCLC cells curcumin inhibited cell migration, invasion and angiogenesis by inhibiting JAK/STAT3 signaling activated in response to IL-6. As a consequence, curcumin downregulated the expression of ICAM, VEGF, MMP2 and MMP7 STAT3-regulated genes involved in tumor invasion [110]. Curcumin inhibited the JAK/STAT3 signaling pathway also in SKOV3 human ovarian cancer cells, causing a decrease in fascin, an actin-binding protein involved in cell adhesion, migration, and invasion [15]. Curcumin is able to prevent invasion by inhibiting AKT, mTOR and P70S6K phosphorylation, as demonstrated in human melanoma A375 and C8161 cells [72] and TC1889 human thymic carcinoma cells [111]. Additionally, it has been demonstrated that curcumin reduced cell invasion and migration in NSCLC A549 cells by increasing miR-206, which further suppressed PI3K/AKT/mTOR pathway activation [112]. The observation that curcumin inhibited NEDD4-mediated signaling in SNB19 and A1207 glioma cells, thus interfering with cell motility, is very interesting [113]. NEDD4 is a E3-ubiquitin ligase involved in the degradation of CNrasGEFs, guanine nucleotide exchange factors (GEFs), that serve as RAS activators, thus, promoting glioma cell migration and invasion [114].

## 3. Bioavailability of Curcumin and Therapeutic Promises

Although the beneficial effects of curcumin are known, it has not yet been approved as a therapeutic agent due to its low bioavailability [21]. Among factors contributing to this limit, the following can be included: low water solubility, poor absorption, low tissue distribution, high rate of metabolism, inactivity of metabolic products and/or rapid elimination and clearance from the body [115]. Curcumin undergoes extensive phase I and II biotransformation [116]. The primary site of metabolism for curcumin is the liver, together with the intestine and gut microbiota; curcumin double bonds are reduced in enterocytes and hepatocytes by a reductase to produce dihydrocurcumin, tetrahydrocurcumin, hexa-hydrocurcumin and octahydrocurcumin. Phase II metabolism that occurs in the intestinal and hepatic cytosol is quite active on both curcumin and its phase I metabolites, especially through conjugation reaction with glucuronic acid and sulfate at the phenolic site catalyzed by UGTs and SULTs enzymes, respectively [116].

Over the years, in order to improve curcumin pharmacokinetic profile and cellular uptake, several strategies have been developed. These include curcumin structural derivatives, analogues preparation and novel drug delivery systems that could enhance its solubility and extend its plasma residence time.

### 3.1. Curcumin Structural Derivatives and Analogues

Structural modifications on the curcumin chemical backbone led to curcumin derivatives and analogues [22,23,24,117,118]. The derivatives category includes all those compounds that maintain the basic structure of pharmacophore and, specifically, the two phenolic rings and the α, β-unsaturated dichetonic bridge which are the portions responsible for the molecule pharmacological activities (Figure 2) [119]. Curcumin derivatives are generally synthesized by modification of the hydroxyl group of the phenol ring, which can be acylated, alkylated, glycosylated and aminoacylated (Figure 2A). Studies on the kinetic stability of synthetic curcumin derivatives showed that glycosylation of the pharmacophore aromatic ring improves the water solubility of the compound, which increases its kinetic stability and leads to a better therapeutic response. In the ΙΙ phase of curcumin metabolism, conjugation reactions take place on the hydroxyl groups (4-OH) attached to the phenyl rings of curcumin [116]. Thus, curcumin stability can be increased by masking the 4-OH groups, thereby extending the active molecule retention time in the body. Benzyl rings are also crucial for inhibiting tumor growth; their modification with hydrophobic substituents such as CH3 groups have been linked with an enhancement of the curcumin derivatives antitumor activity [120]. O-methoxy substitution was found to be more effective in suppressing the NF-kB activity, although this modification simultaneously affected curcumin lipophilicity [2]. Methoxy groups could be demethylated to hydroxyl groups. The reactive chain methylene group, responsible for conformational flexibility, is important for its antitumor/anticancer activity but not for redox regulatory or apoptotic activities. This group could be acylated, alkylated or substituted with an arylidene group (Ar-CH), thereby introducing substituents on the C7 chain [121]. The hydrogenation reaction of double bonds and carbonyl groups on the C7 chain allows the simplest derivatives to be obtained (Figure 2B), such as DHC, THC, HHC and OHC [119]. A comparative study on curcumin and its derivatives demonstrated greater antioxidant activity for several hydrogenated curcumin derivatives compared to the original compound [122]. Tetrahydrocurcumin, a non-electrophilic curcumin derivative, showed greater antioxidant activity than DHC and unmodified curcumin, although it failed to suppress STAT3 signaling pathway and to induce apoptosis [123]. This is evidence that the electrophilic nature of curcumin is essential for STAT3 signaling pathway inhibition. Other curcumin derivatives also include those obtained by exploiting the reactivity of the central β-diketone with hydrazine (Figure 2C). Such heterocyclizations reactions lead to a masking and stiffening of the central 1,3-diketone 1,3-ketoenol system [124] and, after evaluation of the antioxidant activity of these compounds, it is possible to assert that several of these azoles are better antioxidants than curcumin [125]. Oxidation and cleavage are further possible modifications to the β-diketone functional group, all suitably operated to improve the characteristics of the original pharmacophore (Figure 2C). Additionally, it is possible to adopt another approach which may help to increase the curcumin bioavailability, such as the formation of metal complexes, or coordination compounds, which are adducts formed by the reaction of Lewis acids and bases [126]. Metal complexes are generally obtained by reacting curcumin, which exploits the chelating capacity of the β-diketone group with a metal salt. The metals most used for this purpose are Boron, Copper, Iron, Gallium, Manganese, Palladium, Vanadium, Zinc and Magnesium. By complexing curcumin with metal ions, such as Zn^2+^, Cu^2+^, Mg^2+^, an increase in water/glycerol solubility (1:1) and fair stability to light and heat were observed [22,126].

Numerous curcumin analogues have been synthesized and tested to study their interaction with known biological targets and improve the pharmacological profile of this natural product [24,127]. Some curcumin analogues are not obtained starting from the original molecule, but they are synthesized following a condensation reaction between aryl-aldehydes and acetylacetone; through this biosynthetic route, many curcumin analogues have been obtained. The use of acetylacetone derivatives, bearing substituents on the central carbon, leads to analogues with alkyl substituents on the central carbon of the C7 chain [128]. Another strategy concerns the modification of the carbon atom number that makes up the central C7 chain [128]. A greater antitumor activity than curcumin has been observed with the use of a variety of newly synthesized DAP curcumin analogues. These compounds, which possess two aromatic rings (aryl groups) joined by five carbon atoms, were able to suppress cancer growth through modulation of several factors such as NF-kB, MAPK, STAT, AKT-PTEN [127]. DAPs anticancer effects in different cancer cell lines were summarized by Paulraj and co-workers [127].

A particular scientific interest has been shown towards the EF24 analogue, which displayed a better antitumor activity, a lower toxicity in normal cells, a marked increase in bioavailability and a lower metabolic rate compared to the natural compound [23,24]. In vivo studies have shown that, while dietary curcumin is poorly absorbed through the intestinal tract and therefore does not have a therapeutic effect at low doses, on the contrary, EF24 has greater oral bioavailability in mice [129], explaining, to some extent, its improved in vivo activity compared to curcumin. Several studies indicated that EF24 reduces cancer cell growth by inducing cell cycle arrest followed by caspase-mediated apoptosis [130,131]. These actions occur by modulating multiple pathways that determining the inhibition of NF-kB [132] and HIF-1α activity [133] and regulating reactive oxygen species (ROS) [131,134]. NF-kB signaling suppression has been found to be a fundamental aspect for its anticancer activity, since NF-kB is a transcription factor involved in the regulation of genes that monitor cell proliferation, differentiation, cell cycle control and metastasis [135]. According to a discovery by Yin et al. [136], EF24 is able to inhibit the catalytic activity of the IKK protein complex, which blocks the phosphorylation of IkB and causes its degradation while preventing the nuclear translocation of the p65 subunit. EF24 conferred radiation-induced cell death mainly by inhibiting radiation-induced NF-kB signaling in MCF-7 cells [137]. Similar effects were also observed in human neuroblastoma cells [138]. EF24 also regulates HIF-1α expression, which is closely associated with the outcome of chemotherapy in cancer treatment. EF24 overcomes sorafenib resistance through VHL tumor suppressor-dependent HIF-1α degradation and NF-kB inactivation in hepatocellular carcinoma cells [139]. Another notable EF24 antitumor mechanism is ROS production regulation. Tan et al. showed that EF24 inhibited ROS generation and activated ARE-dependent gene transcription in platinum-sensitive (IGROV1) and platinum-resistant (SK-OV-3) human ovarian cancer cells [134]. On the contrary, the ability of EF24 to increase ROS production and then induce apoptosis in cancer cells via a redox-dependent mechanism was found in breast MDA-MB-231, prostate DU-145 [140], gastric SGC-7901 and BGC-823 [141], and colon HCT-116 and SW-620 human cancer cells [131], suggesting that the EF24 role in ROS induction may be cell type dependent. EF24 ability to interfere with the targeted inhibition of antiapoptotic proteins belonging to the Bcl-2 family is being exploited in clinical settings for the treatment of age-related diseases. Although this mechanism has not been fully elucidated, the function of EF24 and other curcumin analogues as senolytic agents is well demonstrated, as they are able to selectively kill the senescent cells (cells that are no longer able to replicate) that accumulate in various organs and tissues as a result of the progress of the damage that occurs with aging. EF24 exerts its senolytic effect by inducing apoptotic death in target cells via proteasome-mediated downregulation from Bcl-2 family proteins, which represents a protection factor for senescent cells, as they are resistant to the induction of apoptosis as a result of the expression of these proteins [23].

Selvendiran et al. [142] demonstrated the ability of curcumin analog HO-3867 to reduce in vitro and in vivo ovarian cancer growth. In particular, this compound enhanced the therapeutic potential of cisplatin in A2780R drug-resistant ovarian cancer cells. The results confirmed that the co-administration of HO-3867 with cisplatin resulted in greater inhibitory effects than cisplatin alone, with cell cycle arrest and apoptotic mechanism being significantly induced by targeting the STAT3 pathway in both in vitro cells and in vivo xenograft tumors [142].

A recent study investigated the antitumor properties of MS13, another diarylpentanoid curcumin analog, on primary (SW480) and metastatic (SW620) human colon cancer cells [143]. MS13 was more cytotoxic in a dose-dependent manner and had a higher growth inhibitory effect towards SW480 and SW620 cells compared to curcumin. Several factors may explain its increased cytotoxicity activity; these include removal of β-diketone, the reduction in the electron donation capacity of OH at position 4′ [144] or the 3′,4′-dimethoxy or 3′-methoxy-4′-hydroxy substituents on the phenyl rings [145].

Recently, Shen et al. [146] tested the efficacy of the B14 analog in MCF-7 and MDA-MB-2310 breast cancer cells. The results indicated that B14 was more potent than curcumin in inhibiting cell viability, colony formation, migration and invasion. Particularly, this analog, functioning simultaneously in multiple pathways, displayed a selective antitumor activity on MCF-7 and MDA-MB-231 cells, but not on MCF-10A breast epithelial cells. Furthermore, in tumor-bearing mice, analog B14 significantly reduced tumor growth and inhibited cell proliferation and angiogenesis. Additionally, pharmacokinetic tests revealed that B14 was more stable and bioavailable than curcumin in vivo [146].

### 3.2. Curcumin Delivery Systems

As already stated, despite curcumin’s remarkable beneficial biological effects, it showed low water solubility in acid and neutral conditions, chemical instability in neutral and alkaline environments, and rapid enzymatic metabolism, which limit its bioavailability. Curcumin bioavailability can be enhanced by:(a).Delaying metabolism through its entrapment within the hydrophobic phases that isolate it from aqueous phase or cell membranes enzymes;(b).Improving its bioaccessibility through an increase in the quantity that is solubilized inside the mixed micelles present in the small intestine; this can be achieved by inserting surfactants, phospholipids, fatty acids or monoglycerides into the curcumin-loaded carrier particles;(c).Promoting its absorption by loading curcumin into particles carrier that contain substances able to increase epithelium cell membranes permeability or block efflux transporters [147]. Therefore, in order to ameliorate curcumin’s pharmacokinetic characteristics, various methodological approaches have been attempted, such as polymeric approaches, magnetic approaches, solid lipid nanoparticles, liposomes, phytosomes, micelles, β-cyclodextrins and solid dispersions [21,25,26,27,28,29,30,148,149,150] (Figure 3) [151,152,153,154]. In addition to these approaches, curcumin conjugation with substances, such as piperine, which is able to inhibit its metabolism [27,155], has emerged as a prominent solution to increase curcumin serum concentration.

#### 3.2.1. Nanoparticles

Nanotechnology is a fast-developing field that is attracting an increasing amount of attention in the fields of drug delivery and cancer therapy, which provides an important route to develop the aqueous formulations of hydrophobic drugs [156]. Nanoparticles, which are 1000 times smaller than the human cell average, possess unique physical, chemical and biological properties that can be useful for both controlled and targeted drug delivery, and for improving the pharmacokinetics and solubility of drugs [156]. It is important to note that the particle sizes of the neosynthesized carriers may influence the therapeutic effects and drug biocompatibility, as well as the chemical-physical characteristics of the devices used [156]. Various types of nanoparticle, such as polymeric, solid lipid and inorganic nanoparticles [156], are widely used to enhance the therapeutic applications of curcumin. Using nanoparticle formulations, it is possible to increase the curcumin water solubility, ensure its intracellular delivery [157], improve the efficacy and limit the toxicity in the settings of cancer therapy, as well as inducing the chemo and radio sensitization of cancer cells [158].

##### Polymeric Nanoparticles

Polymeric nanoparticles (NPs), which possess the advantage of being small and biocompatible, are prepared using either natural or synthetic biodegradable polymers such as silk fibroin, chitosan, PEG, PLA), PGA, PCL and PLGA [156,157,159] (see Table 2 for a summary of the data).

Formulations of nanoparticles based on polymer PLGA have been shown to be effective for enhancing curcumin’s therapeutic effects against cervical cancer [160]. Specifically, the results provided show that nano-curcumin effectively inhibited the growth of Caski and SiHa cervical cancer cells, arrested the cell cycle in the G1-S transition phase and induced apoptosis. Both cell lines revealed a significantly more evident inhibitory effect of curcumin/nano-curcumin on cell proliferation at higher concentrations (20 and 25 μM) and nano-curcumin was found to be more effective than free curcumin at reducing the clonogenic capacity of cervical cancer cells. Furthermore, treatment with nano-curcumin caused a marked decrease in miRNA-21 levels, an onco-miRNA associated with chemoresistance, in vitro and in vivo models, and improved expression of miRNA-214 tumor suppressor [160]. In another study, the anticancer potential of curcumin-encapsulated PLGA nano-formulation was investigated [161]. This formulation showed a superior cell uptake, retention and release in A2780CP highly metastatic ovarian and MDA-MB-231 breast cancer cells [161]. Additionally, it has a greater antiproliferative effect than free curcumin, as demonstrated by the IC50 values; the IC50 of nano-curcumin is 13.9 µM and 9.1 µM in A2780CP and MDA-MB-231 tumor cells, respectively, while the free curcumin showed higher IC50 values (15.2 µM in A2780CP and 16.4 µM in MDA-MB-231 cells compared to nano-curcumin in metastatic tumor cells [161]. The curcumin PLGA nanoparticles effect on cellular viability have been evaluated in LNCaP, PC3 and DU145 cancer and PWR1E non-tumorigenic prostate cell lines [162]. The results showed that the IC50 of this formulation ranged from 20 μM to 22.5 μM, while that of free curcumin ranged from 32 μM to 34 μM across all tumor cell lines, representing an almost 35% reduction in the IC50 value with curcumin-loaded nanoparticles. The evaluation of molecular mechanism of curcumin PLGA nanoparticles displayed that they were able to arrest cell cycle and induce apopotosis by interfering with the NF-kB activity [162]. To increase the anticancer efficiency of curcumin, Bisht et al. designed curcumin polymeric nanoparticles using the micellar aggregates of cross-linked and random copolymers of NIPAAM, with VP and PEG-A [163]. The authors revealed that these NPs were heavily absorbed and were able to block the clonogenicity of the MiaPaca pancreatic cancer cell lines compared to untreated cells or cells exposed to empty polymeric nanoparticles [163]. The superior anticancer effects of curcumin-loaded nanoparticles prepared with amphilic methoxy poly(ethylene glycol)-polycaprolactone (mPEG–PCL) block copolymers compared to free curcumin was confirmed in a human lung adenocarcinoma A549 transplanted mice model [164]. Moreover, curcumin NPs showed little toxicity to normal tissues including bone marrow, liver and kidney at a therapeutic dose. Curcumin silk fibroin nanoparticles (CUR-SF NPs) provided a more stable release and generated much more emphasized anticancer effects than the results obtained with the curcumin free in HCT116 colon cancer cells [165]. In this study, the methods underlying how the controlled release of CUR-SF NPs is able to improve the curcumin cellular uptake in tumor cells were elucidated. Interestingly, the anticancer effect of CUR-SF NPs that involved cell-cycle arrest in the G0/G1 and G2/M phases and apoptosis induction in cancer cells, was improved, while the side effect on normal human colon mucosal epithelial cells was reduced [165]. In another work, curcumin- and piperine-loaded zein-chitosan nanoparticles were characterized by investigating their shapes, morphologies, particle sizes and cell cytotoxicity [166]. The results showed that zein-chitosan nanoparticles loaded with curcumin and piperine have an average size of about 500 nm and high encapsulation efficiencies for curcumin (89%) and piperine (87%). Furthermore, this formulation showed good cytotoxic effects on SH-SY5Y neuroblastoma cell line [166]. A recent study confirmed the enhanced solubility and bioavailability of curcumin within chitosan nanoparticles [167]. This formulation exhibited a sustained release of drug from NPs and a four-fold higher cytotoxic activity on HeLa cervical cancer cells; in fact, DNA damage, cell cycle blockage and elevated ROS levels confirmed the anticancer activity following apoptotic pathways caused by this formulation.

##### Solid Lipid Nanoparticles

Solid lipid nanoparticles (SLNs) are colloidal lipid carriers of size between 50 and 1000 nm and composed by biodegradable physiological lipids [168]. Unlike liposomes, they are rigid particles used both for hydrophobic drugs loading and for their controlled and targeted delivery to the reticuloendothelial system. SLNs possess the advantages of high drug load capacity, good stability, excellent biocompatibility and increased bioavailability [169]. In SLN preparation, lipids are used with a low melting point and solids at room or body temperature and surfactants through various methods including HPH [170]. Among the solid lipids used in SLN preparation there are monostearin, glyceryl monostearate, precyrene ATO 5 (mono, di, triglycerides of C16-C18 fatty acids), compritol ATO 888, stearic acid and glyceryl trioleate which can improve the curcumin chemical stability [168]. Surfactants such as poloxamer 188, Tween 80 and DDAB, are able to reduce interfacial tension between lipid hydrophobic surface and aqueous environment by acting as surface stabilizers [168]. Several studies have evaluated not only curcumin SLNs physico-chemical properties, stability, bioenhancement, bioavailability, but also cellular uptake in vitro and antitumor efficacy in vitro and in vivo [171,172,173,174] (see Table 2 for a summary of the data). Curcumin SLNs using tristearin and PEGylated surfactants have recently been prepared [174]. The study showed that SLN-loaded curcumin with long PEGylates showed increased absorption and long-term stability after oral administration in rats. Furthermore, the bioavailability of curcumin was also 12 times higher in SLNs formulated with long PEGylates than in those formulated with shorter PEGylates [174]. In another study, it has been demonstrated that the presence of sodium caseinate (NaCas) and sodium caseinate-lactose (NaCas-Lac) conjugates as bioemulsifiers to stabilize curcumin SLNs provided a steric hindrance, allowing dispersibility and greater curcumin stability at pH acid. In addition, this formulation displayed a better antioxidant activity compared to that of free curcumin [175]. However, the SLNs use as oral delivery system is limited by drug burst release from SLNs in acid environment. In order to inhibit curcumin rapid release in acid conditions and improve curcumin bioavailability, Baek et al. prepared curcumin SLNs coated with N-carboxymethyl chitosan [171]. The results showed that this formulation showed a prolonged release in simulated intestinal fluid and greater absorption and oral bioavailability compared to free curcumin. Furthermore, curcumin exhibited a strong cytotoxicity compared to curcumin solution in MCF-7 breast cancer cells [171]. Using glyceryl monostearate and poloxamer 188 surfactant were developed curcumin SLNs and evaluated its efficiency in MDA-MB-231 breast cancer cells [176]. The results confirmed that curcumin was stably encapsulated in the lipid matrix and its solubility and release were improved compared to the free curcumin solution. Moreover, curcumin SLNs exhibited higher cellular uptake and higher cytotoxicity by apoptosis induction compared to the free drug [176]. Guorgui et al. demonstrated that plasma levels of curcumin encapsulated in SLNs and in d-α-Tocopheryl polyethylene glycol 1000 succinate-stabilized curcumin (TPGS-CUR) increased when administrated in mice [172]. Additionally, these formulations reduced growth of Hodgkin’s lymphoma xenograft models by 50.5% and 43.0%, respectively, compared to free curcumin [172]. Similarly, inhibitory effects were observed in Hodgkin lymphoma L-540 cells, as proven by curcumin SLNs ability to reduce the expression of XIAP and Mcl-1, proteins involved in cell proliferation and apoptosis, and of cytokines IL-6 and TNF-α as well as to enhance the growth inhibitory effect of bleomycin, doxorubicin and vinblastine chemotherapeutic drugs [172].

##### Inorganic Nanoparticles

Recently, inorganic nanoparticles have received considerable attention, particularly in the oncology field for diagnostic and therapeutic applications [177,178]. They have specific physical size-dependent properties, such as the contrasting effect and magnetism as well as good microbial resistance and excellent storage capacity [179].

The in vitro and in vivo therapeutic efficacy of curcumin-loaded magnetic nanoparticles (MNPs) was evaluated in several tumors including pancreatic cancer [180] (see Table 2 for a summary of the data). Specifically, HPAF-II and Panc-1 human pancreatic cancer cells exhibited efficient internalization of this formulation in a dose-dependent manner. Moreover, they inhibited both in vitro HPAF-II and Panc-1 cell proliferation and in vivo tumor growth in an HPAF-II xenograft mouse model. The growth-inhibitory effect of MNPs-CUR formulation correlated with the suppression of PCNA, Bcl-xL, induced Mcl-1, MUC1, collagen I, and enhanced membrane β-catenin expression. Interestingly, MNPs-CUR formulation improved serum bioavailability of curcumin in mice up to 2.5-fold as compared with free curcumin [180]. Folic-acid-tagged aminated-starch-/ZnO-coated iron oxide nanoparticles were prepared for the targeted delivery of curcumin. Particularly, the authors provided evidence that ZnO-incorporated aminated-starch-coated iron oxide nanoparticles showed a significant controlled release of curcumin in vitro and reduced HepG2 liver and MCF-7 breast cancer cell viability without toxic effects on human lymphocytes [181]. In addition, folic acid taging to the nanoparticles led to both an increase in cellular uptake and in ROS generation [181]. The curcumin pluronic stabilized Fe3O4 magnetic nanoparticles (CUR-PSMNPs) showed aqueous colloidal stability, biocompatibility as well high loading affinity for curcumin and better curcumin release in acidic conditions [182]. Additionally, cell viability studies confirmed that curcumin and CUR–PSMNPs significantly reduced MCF-7 cell proliferation with IC50 values of 25.1 and 18.4 μM, respectively. The higher toxicity of CUR–PSMNPs were due to its significant greater cellular uptake respect to pure drug [182]. PEGylated curcumin was used as the surface modification of magnetic nanoparticles (MNP@PEG-CUR) in order to simultaneously ameliorate magnetic targeting characteristic of nanoparticles and PEG conjugated drug. The results indicated that MNP@PEG-CUR showed higher drug release in acidic conditions, biocompatibility and low cytotoxicity at physiological pH [183]. Gangwar et al. prepared curcumin conjugated silica coated nanoparticles and observed good stability in aqueous medium, sustained drug release and greater anticancer properties. Particularly, cytotoxicity analysis revealed that conjugate was more toxic on HeLa cervical cancer cells compared to normal fibroblasts [184]. In another work, Manju et al. reported the synthesis of water-soluble HA-CUR@AuNPs attached to curcumin and explored their targeted delivery onto cancer cells (C6 glyoma, HeLa cervical and Caco-2 colon cancer cells). The results indicated that this formulation displayed good aqueous solubility, superior cellular uptake and anticancer effects respect to free curcumin. The authors highlighted how the greater targeting efficacy via hyaluronic acid- and folate receptor-mediated endocytosis could increase the overall intracellular accumulation of HA-CUR@AuNPs [185].

#### 3.2.2. Liposomes

Liposomes are self-assembled spherical vesicles of different sizes ranging from 20 nm to several microns in diameter and composed of one or more bilayers surrounding aqueous unit [186]. Liposomes that can carry both hydrophilic and hydrophobic molecules are excellent drug delivery systems characterized by high biocompatibility and biodegradability, high stability, low toxicity, better solubility, targeting for specific cells, controlled distribution, flexibility and ease of preparation [186]. However, in some cases, they can undergo rapid elimination from the bloodstream, physical and chemical instability, aggregation, fusion, degradation, hydrolysis and phospholipid oxidation [187]. The liposomal systems are designed with the aim of allowing curcumin to be distributed over aqueous medium and increasing its bioavailability and therapeutic effect [188] (see Table 2 for a summary of the data). The preparation method used affects the nature of the formed liposomes, as well as curcumin-carrying capacity. One study compared the characteristics of curcumin-loaded liposomes prepared using three different methods such as thin film, ethanol injection and pH-based methods [189]. The results indicated that both liposomes initial diameters and encapsulation efficiency decreased with the following trend, respectively: thin film (453 nm; 78%)> pH-driven (217 nm; 66%)> injection of ethanol (115 nm; 39%). Furthermore, it has also been shown that the source of phospholipids can also affect the physicochemical characteristics of curcumin-loaded liposomes. Liposomes produced from milk fat globule membranes (MFGMs) offered better protection than those of soy lecithin [190]. However, the instability of conventional liposomes under the physiological conditions found in the gastrointestinal tract hindered clinical applications. Indeed, coated liposomes with thiolated chitosan or other agents are made in order to improve liposome stability and then curcumin bioavailability [187,191]. It has been observed that liposome formulations improved curcumin aqueous solubility and bioavailability by 700-fold and to 8–20-fold, respectively, in tumor-bearing mice [192]. Moreover, antiproliferative activity of DMPC liposomal curcumin was evaluated in LNCaP and C4-2B prostate cancer cell lines [193]. The results showed that liposomal curcumin caused a greater inhibition of cell proliferation than free curcumin and even at doses 10 times lower [193]. Dhule et al. showed that the 2-hydroxypropyl-γ-cyclodextrin/curcumin-liposome complex was able to inhibit KHOS osteosarcoma cancer cell proliferation by inducing apoptosis in vitro as well as reduce the growth of a xenograft osteosarcoma model in vivo [194]. Furthermore, liposomes encapsulating doxorubicin and curcumin, reduced C26 colon cancer cell proliferation to a greater extent than those loaded with doxorubicin alone [195]. A liposomal curcumin formulation also resulted in the dose-dependent inhibition of the proliferation and motility and apoptosis induction in Ishikawa and HEC-1 endometrial cancer cells. These inhibitory effects occurred probably through negative regulation of the NF-kB pathway [196]. Moreover, a recent study demonstrated that small curcumin molecules encapsulated in liposome nanocarriers significantly increased the blue-light-emitting diode (BLED)-induced photodynamic therapy (BLED-PDT) effect, by increasing intracellular ROS levels and apoptosis in human lung A549 cancer cells [197].

#### 3.2.3. Phytosomes

Curcumin is also often complexed with different types of phospholipids such as phosphatidylcholine, or other lipid carriers, in order to create phytosomal formulations. The use of phosphatidylcholine in the molecule complexation is useful in allowing the efficient transport across cell membranes and then absorption. In the phytosome, it is possible to observe an interaction between the active substance (i.e., curcumin) and the polar head of the phospholipid; this bond permits curcumin to become an integral part of the membrane [198]. Phytosomes application for the curcumin delivery has the advantage of improved its absorption and bioavailability as well as enhanced its therapeutic benefits [199,200,201] (see Table 2 for a summary of the data). The relative absorption of standardized curcuminoid mixture (curcumin, demethoxycurcumin, and bisdemethoxycurcumin) and its corresponding lecithin formulation (Meriva) was investigated in a randomized, double-blind, crossover human study. The results indicated that total curcuminoid absorption was about 29-fold higher for Meriva compared to unformulated curcuminoid mixture. Therefore, the improved curcuminoids absorption could explain the Meriva clinical efficacy at significantly lower doses than unformulated curcuminoids [202]. A recent study, which a CPC was prepared and solidified with Soluplus [polyvinyl caprolactam-polyvinyl acetate polyethylene glycol graft copolymer] carrier, demonstrated that solydifing process enhanced CPC flowability and dissolution rate as well as curcumin oral bioavailabilty in rats [203].

#### 3.2.4. Micelles

Micelles consist of self-assembled aggregates of surfactants or block copolymers ranging in size from 10 to 100 nm [204]; they can be formed by dissolution, dialysis, emulsion, solvent evaporation and lyophilization [204]. Micelles, given their small molecular size that ensure an effective macromolecules transport across cell membrane, are prepared to solubilize curcumin [205]. Micelles, are systems capable to improve the gastro-intestinal absorption of curcumin, increasing plasma levels and decreasing the kinetics of elimination, with a consequent enhancement in bioavailability and a hepato-protective effect likened to free curcumin [205] (see Table 2 for a summary of the data). In addition, an increase in water solubility facilitates the development of a stable and homogenous solution for intravenous applications. Additionally, the nanoscale of micelles and presentation of the hydrophilic stabilizing interface can prolong their circulation time in vivo and enhance the cellular uptake [205]. Letchford et al. formulated curcumin micelles that contained copolymers of deblock MePEG-b-PCL that achieved an approximately 13 × 10^5^-fold increase in water solubility [206]. Similarly, a previous study confirmed a 60-fold increase in the biological half-life of curcumin polymeric micellar compared to the free form when administered orally in rat models [207]. Schiborr et al. also observed that the use of curcumin micelles in volunteers determined a significant increase in drug plasma concentration formulated in micelles compared to that of native curcuminoids [208]. The uptake and transepithelial transport of native curcumin and micellar formulation (Sol-CUR) have been evaluated using a Caco-2 cell model [209]. The results showed that Sol-CUR caused a 185-fold increase in AUC compared to native curcumin. This curcumin oral bioavailability improvement was dependent on Sol-CUR’s higher intestinal absorption and distribution [209]. Recently, pharmacokinetic studies confirmed a significant enhancement of aqueous solubility, as well as stability, dissolution, and permeability of curcumin formulated in micelles compared to free drug [210]. A study by Chen et al. focused on the synthesis, physico-chemical characterization and in vitro evaluation of curcumin loaded in the super hydrophilic zwitterionic polymers PSBMA micelles. The results demonstrated that this innovative formulation exhibited greater stability, cellular uptake and tumor cytotoxicity compared to native curcumin [211]. Interestingly, various studies using curcumin polymer micelles may provide an indication of very promising therapeutic potential in cancer treatment. Curcumin delivery from nano-sized polymer micelles formulated with MPEG-P [CL-co-PDO] were evaluated in PC-3 human prostate cancer cell line. The results indicated that mixed micelle copolymers possessed a higher encapsulation efficiency (>95%), a prolonged drug release profile, and a dose-dependent cytotoxicity effect on tumor cells in contrast to that of free curcumin [212]. Patil et al. prepared CUR-MM of Pluronic F-127 (PF127) and Gelucire^®^ 44/14 (GL44) in order to enhance its oral bioavailability and cytotoxicity in A549 human lung cancer cell line [213]. The in vitro dissolution profile of CUR-MMs revealed a controlled curcumin release. Furthermore, this formulation showed a significant improvement in in vitro cytotoxic activity and in vivo oral bioavailability of curcumin by approximately 3 and 55 times, respectively, compared to curcumin alone. These effects could be attributed to solubilization of hydrophobic curcumin into micelle core as well as to the ability of PF127 and GL44 to inhibit P-gp mediated efflux. In another work, it has been demonstrated that CUR-MPP-TPGS-MMs showed small size, high drug-loading and sustained release profile. Furthermore, they improved the intestinal absorption of curcumin after oral administration and then oral bioavailability in rats [214].

#### 3.2.5. Curcumin/β-Cyclodextrin and Solid Dispersions Formulations

Among various strategies performed to improve oral curcumin solubility there are cyclodextrin (CD) inclusion complexes and solid dispersions (SDs) [215] (see Table 2 for a summary of the data). Cyclodextrins are bucket-shaped oligosaccharides, consisting of six (a-), seven (b-) or eight (g-) units of D-glucopyranose linked through an a-1,4-glycosidic bond to form macrocycles and are widely known as solubilizing and stabilizing agents [216]. β-CD, with its hydrophilic outer surface and hydrophobic inner cavity, has become a benchmark for nanotechnology research due to its modern lumen size, high drug load, low cost of production, best stability of the lipophilic drug, easy modification by molecules and good biocompatibility characteristics [217]. Mangolim et al. demonstrated that curcumin-β-CD complex exhibited a sunlight stability 18% higher and a 31-fold increased solubility compared to the pure drug [218]. Liquid-type β-cyclodextrins curcumin delivery carrier is one of the most widely used types in the food industry [219]. Particularly, the delivery carrier of hydroxypropyl β-CD curcumin prepared by saturated aqueous solution increased curcumin solubility in water by 276 times [220] and oral bioavailability by 3 times [221]. In addition, the solid granule-based β-cyclodextrins curcumin delivery vector had very good storage stability and increased its bioavailability significantly [222]. The formulations of CD combined with curcumin (CUR-CD) have been shown to have enhanced the antiproliferative, anti-inflammatory and anticancer effects of the nutraceutical agent [22]. The therapeutic effect of CUR-CD encapsulated into positively charged biodegradable chitosan (CUR-CD-CS) nanoparticles was investigated on the SCC25 skin cancer cell line. The study demonstrated that CD presence not only increased curcumin solubility and cellular absorption, but also promoted cell cytotoxicity [223]. Yallapu et al. showed that β-CD-curcumin inclusion complex (CD30) possessed an improved uptake and a greater potent therapeutic efficacy in DU145 prostate cancer cells compared to the free drug [154]. Similarly, the formulation of β-cyclodextrin-curcumin complex (CD15) enhanced curcumin delivery and improved its therapeutic efficacy compared to free curcumin in in vitro A549, NCI-H446 and NCI-H520 human lung carcinoma cell lines and in vivo mouse hepatoma H22 xenograft models. Particularly, through regulation of MAPK/NF-kB pathway, CD15 upregulated p53/p21 pathway, downregulated Cyclin E-CDK2 combination and increased Bax/caspase 3 expression to induce cellar apoptosis and G1-phase arrest [224]. A study demonstrated that in HeLa cervical cancer cells, curcumin encapsulated in crosslinked β-CD nanoparticles acted extremely rapidly on cell metabolism resulting in a significant cancer cell growth inhibition [225]. Recently, a nano-drug system namely FA-CUR-NPs consisting of FA, β-CD, εCL and curcumin was performed to improve curcumin delivery in cervical cancer tissues which overexpress FRs and to achieve controllable release in vitro and in vivo. In this system, FA binding to FRs was used as a targeting molecule, β-CD modified by ε-CL as delivery carrier and for controlling drug release and curcumin as a model drug to limit multidrug resistance after administration [226]. The study demonstrated that in vitro curcumin release rate from FA-CUR-NPs under tumor microenvironment conditions (pH 6.4), was three times faster than that under systemic circulation conditions (pH 7.4). Additionally, in vitro cytotoxicity was proportional to cellular uptake efficiency and the in vivo marked accumulation in tumor site was responsible of greater antitumor activity. These findings indicated FA-CUR-NPs could represent a promising approach for improving cancer therapy through active targeting and controllable release [226]. Moreover, in another work the use of hydroxypropyl-β-CD as a carrier-solubilizer improved solubility of the curcumin–piperine system, its permeability through biological membranes (gastrointestinal tract, blood–brain barrier), the antioxidant and antimicrobial activities and as well as the enzymatic inhibition against acetylcholinesterase and butyrylcholinesterase [227].

The SDs technology transforms crystalline materials into amorphous materials: an active substance is incorporated into a carrier, which is generally selected from suitable polymers [228]. The active ingredient, originally crystalline, is usually transformed into amorphous form in the dispersion and it is for these reasons that SD is a consolidated method applied to improve the solubility and bioavailability of drugs with poor water solubility such as curcumin [228]. The major outcomes of this technique include prolonged survival, antitumor and antimetastasis, anti-inflammatory, antibacterial activity, enhanced stability and bioavailability [229,230]. Seo et al. prepared solid dispersions of curcumin with Solutol^®^ HS15 SD showing greater solubility and bioavailability compared to free curcumin [231]. Similarly, curcumin SDs with cellulose acetate and mannitol showed a better solubility in water and an improvement in oral bioavailability of about seven times compared to curcumin [232]. In another work, Texeira et al. demonstrated how solid dispersions of Gelucire^®^50/13-Aerosil curcumin possess stability over time up to 9 months and an improvement in water solubility and in dissolution rate of about 3600- and 7.3-fold, respectively. Furthermore, the major curcumin gastrointestinal absorption resulted in a systemic bioavailability increase and greater anti-inflammatory activity in rats [229]. Antioxidant and antigenotoxic effects of curcumin formulated in SDs compared to unmodified drug were evaluated in Wistar rats. The results showed that curcumin SDs, even if they did not alter the antigenotoxic effects observed with free curcumin, displayed a better water solubility, with a maximum of absorption in the gastrointestinal tract [233]. Recently, curcumin SDs were obtained using Poloxamer 407 as the encapsulant agent and their cytotoxic effects were evaluated against tumoral (breast, lung, cervical and hepatocellular carcinoma) and PLP2 non-tumoral cells. These formulations that were readily dispersible in water had cytotoxicity against all the tested tumor cell lines without toxic effects for non-tumor cells [234].

#### 3.2.6. Curcumin Conjugates Formulations

Combinations of curcumin with other compounds increase its solubility and cellular absorption, extend the residence in plasma improving the pharmacokinetic profile and then oral bioavailability (see Table 2 for a summary of the data). Curcumin conjugation by covalent bond with piperine, an alkaloid of black pepper and known as an inhibitor of hepatic and intestinal glucuronidation [235], enhanced curcumin serum level and a significantly reduced elimination half-life and clearance, producing 154% and 2000% increase in bioavailability in rats and humans, respectively [236]. Another study also showed that piperine (20 mg/kg orally) when administrated with curcumin (2 g/kg orally) increased the bioavailability of the latter up to 20 times more in epileptic rats [237]. Similarly, both intestinal absorption and curcumin bioavailability has been found to be improved following concomitantly oral intake with piperine in rats [238]. In a clinical study, the curcumin-lecithin-piperine formulation, namely BCM-95CG (Biocurcumax^®^), administrated to healthy subjects aged 28–50 years significantly improved curcumin bioavailability and showed a better pharmacokinetic profile than pure curcumin [239]. An in vitro study conducted by Sing et al. demonstrated that conjugates of curcumin, piperic acid and glycine, prepared by esterifying the sites of metabolic conjugation 4 and 4′ phenolic hydroxyls, triggered a mitochondrion-dependent apoptosis in MCF-7 and MDA-MB-231 breast cancer cell lines; however, the authors highlighted that the conjugation did not affect the efficacy of parent molecule, while these compounds could work in vivo as prodrugs with improved pharmacokinetic profile. This suggested other in vivo studies were needed to elucidate the therapeutic potential of curcumin conjugates in breast cancer [240].

## 4. In Situ Implant Systems

Unlike blood circulation nanosystems that are injected directly into blood vessels, nanoimplant systems have been used as large containers of drugs that must be released directly and exclusively at the application site for the treatment of topical diseases. When loaded into a membrane made with biodegradable nanofibers, curcumin may be helpful in preventing post-operative relapse of solid tumor. Different researchers have attempted to charge curcumin into nanofiber membranes by electrospinning and, by adjusting the curcumin dosage, they have found that it is possible to control the drug’s loading capacity [241]. Curcumin-loaded nanofiber membrane showed a good local delivery of hydrophobic drug, sustained release, biocompatibility and biodegradability and strong cytotoxicity against rat glioma 9L cells [242]. Therefore, this formulation could represent an ideal candidate for the prevention of postoperative tumor relapse after excision [243]. Further to the application of nanocrystallized curcumin in cancer therapy, tissue engineering is another biomedical field where curcumin could also have a particular impact on bone formation. The choice of hydrogels as the base material could be appropriate due to their porosity and practical operation in clinical application. The hybrid hydrogel-micelle system was used to control malignant pleural effusion, which reduced the number of pleural tumor foci and prolonged survival time to malignant pleural effusion of carrier mice [244]. Angiogenesis is also effectively inhibited using this approach, and the micelle-hydrogel hybrid has been shown to be an exceptional transport system.

## 5. Clinical Trials with Curcumin

In recent decades, several clinical studies have been performed to evaluate the effect of curcumin in cancer patients. Depending on the tumor type, curcumin can act by causing a decrease in the patients symptoms or, in other cases, it can lead to an improvement in tumor markers and vital parameters [245]. Currently, 71 clinical studies reported in cancer patients treated with curcumin alone or in combination with other compounds were listed in the United States National Library of Medicine (clinicaltrials.gov). Pharmacologically, curcumin is shown to be well-tolerated and relatively safe to use in patients. The clinical trials conducted, thus far, have reported relatively no toxicity [246]. Phase I clinical trial conducted by Cheng et al. showed that oral administration of 8 g/day of curcumin for 3 months is non-toxic to patients with high-risk or pre-malignant lesions [247].

A phase II trial of curcumin conducted in twenty-five patients with advanced pancreatic cancer showed that oral curcumin was tolerated without toxicity at doses of 8 g/d for up to 18 months [248]. Particularly, it has been observed that two of them showed clinical biological activity; one patient reported having a stable disease for up to 18 months and another had a marked tumor regression along with an increase in the serum levels of cytokines (IL-6, IL-8, IL-10, and IL-1 receptor antagonists) [248]. Additionally, curcumin downregulated expression of the NF-kB, COX-2, and phosphorylated signal transducer and activator of transcription 3 in peripheral blood mononuclear cells from patients [248]. In another phase II pilot study, a combination of docetaxel, prednisone and curcumin was well-tolerated in patients with castration-resistant prostate cancer [249]. Dützmann et al. investigated the intratumoral concentrations and clinical tolerance of micellar curcuminoids composed by 57.4 mg curcumin, 11.2 mg demethoxycurcumin, and 1.4 mg bis-demethoxycurcumin administrated three times day for 4 days in glioblastoma patients [250]. The results revealed that oral administration of this formulation generated quantifiable concentrations of total curcuminoids in glioblastomas with likely effects on intratumoral energy metabolism [250]. Furthermore, in patients with orbital pseudotumors, head and neck squamous carcinoma, breast, lung and prostate cancers, curcumin application demonstrated beneficial effects with reductions in tumors volume and tumor markers [245]. Another study carried out on 11 volunteer patients with osteosarcoma aimed to quantitatively evaluate curcumin levels in the bloodstream following the intake of SLNs; this study highlighted an improvement in the bioavailability of this polyphenol, when compared with the results obtained on subjects who received unformulated curcuminoids extract [173]. However, further studies are needed to evaluate both the long-term tolerability after chronic administration and the relationship between plasma curcumin levels and disease markers. In a phase I clinical trial, an oral formulation of curcumin was evaluated in fifteen patients with advanced colorectal cancer refractory to standard chemotherapies. The results showed an absence of toxicity with curcumin, while 2 of the 15 patients showed stable disease after 2 months of curcumin treatment [251]. Carroll et al., in a phase II clinical study conducted on patients with colon cancer lesions, demonstrated that curcumin administration for 30 days determined a significant 40% reduction in aberrant crypt foci (ACF) [252]. These data suggested curcumin use as a cancer prevention agent. However, although curcumin in cancer patients has often improved life quality and reduced tumor markers, it is also true that curcumin has sometimes shown limited effect in some patients with disease advanced stage. Preclinical studies conducted so far confirmed the important role of curcumin and, in particular, the benefits that its synthetic derivatives or nanoformulations could bring to human health. New efforts are needed to confirm the efficacy of curcumin-based products for the treatment of human diseases and to ensure that the products are non-toxic once introduced into the body. Several ongoing clinical trials should provide a deeper understanding of curcumin-based formulation efficacy and mechanism of action against human diseases.

## 6. Conclusions

Curcumin is one of the most promising clinical compounds of the last few decades. Its therapeutic benefits have been demonstrated in various chronic diseases and, above all, in various cancers. Precisely in the antitumor field, curcumin is able to modulate the action of growth factors, cytokines, transcription factors and genes that regulate cell proliferation, apoptosis and the metastatic process. Some of curcumin’s multiple pharmacological activities have been used experimentally and clinically in both humans and animals. Notable among these are the antioxidant, anti-inflammatory and anticarcinogenic properties, which all seem related. It is encouraging that curcumin is of low toxicity. Most of the data supporting the antitumour activity of curcumin was obtained in vitro. Unfortunately, the challenges of low solubility and rapid elimination and then poor bioavailability, have delayed its adoption as a therapeutic agent. These efforts have led to the development of several curcumin formulations that have been systematically prepared to improve the absorption, water solubility, distribution and, hence, the bioavailability of curcumin. These methods have been promising with respect to preclinical and clinical efficacy and involve the use of both synthetic derivatives and analogues of curcumin as well as formulations of nanoparticles, liposomes, phytosomes, micelles and natural adjuvants such as piperine. All of these approaches generated a significant improvement in the bioavailability, absorption and retention time of curcumin, along with increased delivery to target tissues, as widely reported. However, further investigations are needed to fulfil the significant promise they currently hold and reveal new perspectives for the further enhancement of the therapeutic capacity of this interesting natural molecule. In particular, more epidemiological and clinical trials involving large numbers of subjects should be conducted to support to the obtained in vivo and in vitro results and to confirm curcumin’s usefulness in cancer treatment and/or prevention.

## Figures and Tables

**Figure 1 biomedicines-09-01476-f001:**
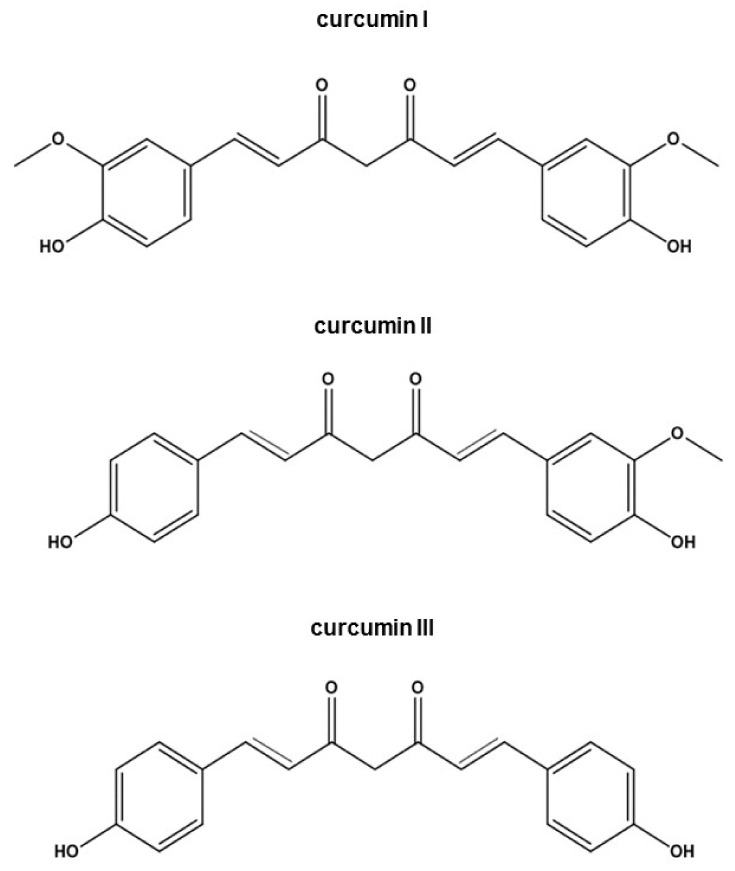
Chemical structures of curcumin I, II, III.

**Figure 2 biomedicines-09-01476-f002:**
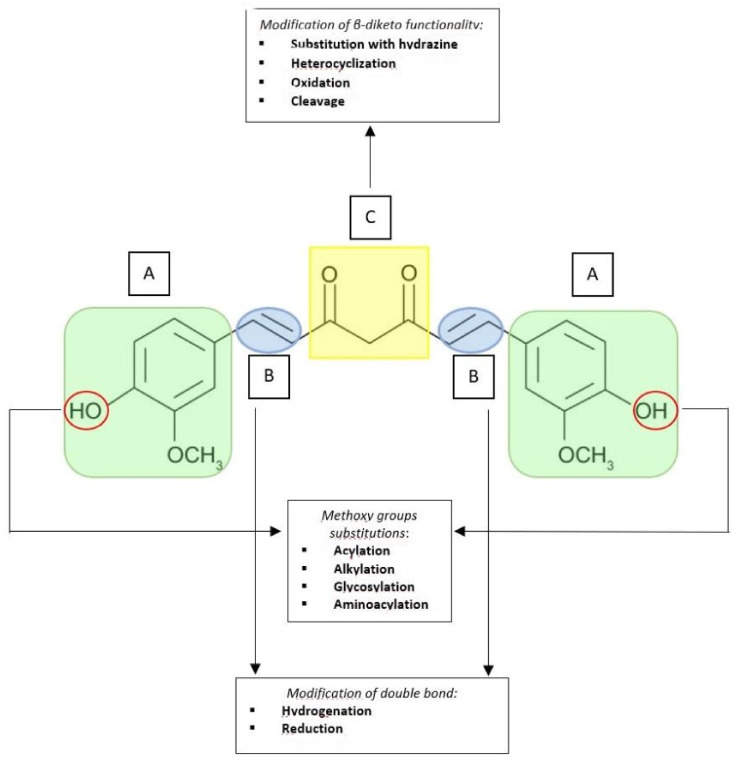
General chemical structure of curcumin derivatives. Curcumin chemical structure include two aromatic rings (**A**) linked to a β-dichetonic group (**C**) through a double bond (**B**).

**Figure 3 biomedicines-09-01476-f003:**
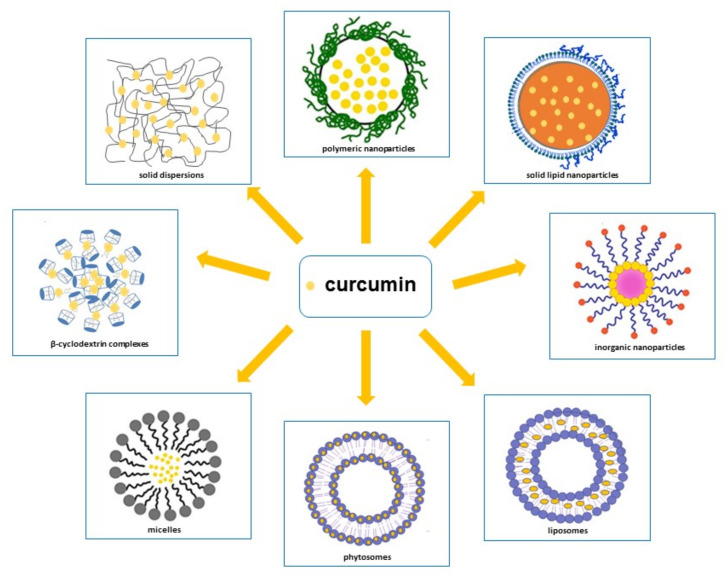
Principal delivery systems to enhance curcumin oral bioavailability. In the figure the polymeric nanoparticles, inorganic nanoparticles and micelles images were adapted from Praditya et al. [151], the phytosomes and liposomes images were adapted from Yang et al. [152], the solid lipid na-noparticles image was adapted from Li et al. [153] and the β cyclodextrin complexes image was adapted from Yallapu et al. [154].

**Table 1 biomedicines-09-01476-t001:** Antitumoral activities of curcumin.

Biological Effects	Mechanisms of action	Cancer type	References
Antiproliferative			
	CDK2 decrease	Breast	[35]
	CDK4 decrease		[34,35]
	Cell cycle arrest at G1 phase		[16,35,36]
	Cell cycle arrest at G2/M phase		[37]
	Cell viability decrease		[39]
	Cyclin A decrease		[35]
	Cyclin D1 decrease		[34,35,37]
	Cyclin E decrease		[35,36]
	DLC1 increase		[40]
	EZH2 decrease		[40]
	GSK3β increase		[37]
	Increased sensibility to chemotherapeutic agents		[39]
	IκB increase		[16]
	NF-kB p65 decrease		[16]
	p21 and p27 increase		[35,36]
	p53 increase		[36]
	β-catenin decrease		[37]
	CDK2 decrease	Colon	[45]
	Cell cycle arrest at G1 phase		[45]
	Cell cycle arrest at G2/M phase		[47,48]
	Cell viability decrease		[44]
	Cyclin A decrease		[46]
	E2F4, decrease		[46]
	p21 and p27 increase		[46]
	p53 increase		[47]
	ROS increase		[46]
	Cell cycle arrest at G2/M phase	Bladder	[49]
	Cell viability decrease		[49,50]
	Cyclin E1 decrease		[49]
	p27 increase		[49]
	Trop2 decrease		[49]
	Cell cycle arrest at G2/M phase	Glioma	[53]
	Cyclin D1 decrease		[52]
	Egr-1 increase		[52]
	FoxO1 increase		[53]
	p21 increase		[52]
	Cell cycle arrest at G2/M phase	Glioblastoma	[54]
	HDGF / β-catenin complex inhibition		[56]
	p57 increase		[54]
	Skp2 decrease		[54]
	CDK4 decrease	Gastric	[67]
	Cell cycle arrest at G1 phase		[60]
	Cell cycle arrest at G0/G1-S phase		[67]
	c-myc decrease		[61]
	c-myc/(lncRNA) H19 pathway downregulation		[63]
	Cyclin D1 decrease		[67]
	EGF/PAK1/NF-kB/cyclin D1 pathway inhibition		[60]
	LRP6 ans phospho-LRP6 decrease		[61]
	miR-33b increase		[68]
	miR-34a increase		[67]
	ODC activity decrease		[59]
	p21 increase		[62]
	p53 increase		[62,63]
	PI3K signaling inhibition		[62]
	SMOX mRNA and activity increase		[59]
	Wnt3a decrease		[61]
	XIAP decrease		[68]
	β-catenin and phospho β-catenin decrease		[61]
	Akt/mTOR pathway downregulation	Melanoma	[72]
	Cell cycle arrest at G2/M phase		[70,71,72]
	Cyclin A decrease		[71]
	DNMT1 decrease		[71]
	IKK inhibition		[73]
	iNOS inhibition		[70]
	NF-kB inhibition		[70,73]
	p21 and p27 increase		[71]
	PDE decrease		[71]
	UHRF1 decrease		[71]
**Pro-apoptotic**			
	Bad increase	Breast	[77]
	Bax increase		[79]
	Bax/Bcl-2 ratio increase		[78]
	Bcl-2 decrease		[79,94]
	Cleaved caspase 3 increase		[78]
	Cleaved Parp-1 increase		[77]
	FAS inhibition		[80]
	GSH decrease		[78]
	Histone H3 acetylation and glutathionylation increase		[74]
	miR-15a and miR-16 increase		[94]
	NF-kBp65 decrease		[79]
	p53 increase		[77]
	ROS increase		[78]
	Bcl-2 decrease	Melanoma	[81]
	JAK-2/STAT-3 signaling inhibition		[81]
	p53-independent Fas/caspase 8 pathway activation		[82]
	Akt signaling inhibition	Ovarian	[83]
	Bcl-2 and survivin decrease		[83]
	p38 MAPK activation		[83]
	Bax and p53 increase	Myeloma	[84]
	MDM2 decrease		[84]
	AMPK increase	Colon	[85]
	APC decrease		[87]
	Bax increase		[89,90,91]
	Bcl-2 decrease		[86,89,90,91]
	Bcl-xL decrease		[86]
	Caspase 3 activation		[87,89]
	Caspase 7 activation		[89]
	COX2 decrease		[86]
	Cyclin D1 decrease		[86]
	DR5 upregulation		[88]
	E-cadherin decrease		[87]
	Fas-mediated caspase 8 activation		[89]
	IAP-2 decrease		[86]
	Mithocondrial [Ca^2+^] increase		[89]
	Mithocondrial cytochrome c release		[89]
	Mithocondrial membrane potential reduction		[89]
	pAKT decrease		[85]
	β-catenin decrease		[87]
	Caspase 3/7 activation	Bladder	[50]
	Caspase 3/7 activation	Glioblastoma	[92]
	Bax increase		[92]
	Bcl-2 decrease		[92]
	Caspase 3 activation	Hosteosarcoma	[93]
	Parp-1 cleavage		[93]
	DNA damage	Gastric	[59]
	ODC activity decrease		[59]
	ROS production		[59]
	miR-186 pathway activation	Lung	[95]
	Bax and cleaved caspase 3/9 increase	Bladder	[96]
	Bcl2 decrease		[96]
	miRNA 344a-3p increase		[96]
**Antimetastatic**			
	Axl decrease	Breast	[98]
	CD24 decrease		[98]
	CXCL1 and 2 decrease		[101]
	miR-34a increase		[98]
	NF-kB inhibition		[100,101]
	Rho-A decrease		[98]
	RhoA/ROCK/MMPs/Vimentin pathway inhibition		[103]
	Slug decrease		[98]
	uPA decrease		[100]
	JAK/STAT3 pathway inhibition	Retinoblastoma	[104]
	miR-99a increase		[104]
	MMP2 decrease		[104]
	RhoA decrease		[104]
	ROCK1 decrease		[104]
	Vimentin decrease		[104]
	MMPs signaling pathways inhibition	Bladder	[50]
	Trop2 decrease		[49]
	Cellular matriptase downregulation	Prostate	[106]
	MMP9 decrease		[106]
	Angiogenesis inhibition	Colon	[44]
	Claudin-3 decrease		[107]
	Metastasis inhibition		[44]
	MMP9 decrease		[107]
	MMP2 decrease	Melanoma	[81,108]
	MMP9 decrease		[81]
	NF-kB signaling pathways inhibition		[108]
	TIMP-2 increase		[81]
	NF-kB and Wnt/βcatenin pathways inhibition	Cervical	[17]
	ICAM decrease	SCLC	[110]
	VEGF decrease		[110]
	MMP2 and MMP7 decrease		[110]
	STAT3 decrease		[110]
	IL-6-inducible JAK/STAT3 phosphorylation reduction		[110]
	Fascin decrease	Ovarian	[15]
	JAK/STAT3 signaling pathway inhibition		[15]
	pAkt, pmTOR, pP70S6K downregulation	Melanoma	[72]
	miR-27a decrease	Thymic	[111]
	mTOR and Notch-1 pathways inhibition		[111]
	miR-206 increase	NSCLC	[112]
	PI3K/AKT/mTOR pathway inhibition		[112]
	NEDD4 signaling pathways inhibition	Glioma	[113]

**Table 2 biomedicines-09-01476-t002:** Strategies to enhance bioavailability and/or antitumoral effects of curcumin.

Composition	Outcomes	References
**Polymeric Nanoparticles (NPs)**		
PLGA + CUR	Superior anticancer effects respect to free curcumin in cervical cancer cells	[160]
PLGA + CUR	Superior cellular uptake and anticancer effects respect to free curcumin in ovarian and breast cancer cells	[161]
PLGA + CUR	Superior cellular uptake and anticancer effects respect to free curcumin in prostate cancer cells	[162]
NIPAAM+ VP + PEG-A + CUR	Superior cellular absorbition and anticancer effects respect to free curcumin in pancreatic cancer cells	[163]
mPEG-PCL + CUR	Superior anticancer effects respect to free curcumin in human lung adenocarcinoma cancer cells	[164]
silk fibroin + CUR	Superior cellular uptake and anticancer effects respect to free curcumin in colon cancer cells	[165]
zein-chitosan +CUR	High encapsulation efficiencies for curcumin. Superior anticancer effects respect to free curcumin in neuroblastoma cell line	[166]
chitosan +CUR	Enhanced curcumin solubility and bioavailability. Sustained drug release from NPs and anticancer effects with respect to free curcumin in cervical cancer cells	[167]
**Solid Lipid Nanoparticles (SLNs)**		
SLNs + N-carboxymethyl chitosan+ CUR	Prolonged release in simulated intestinal fluid, greater absorption and oral bioavailability compared to free curcumin. Superior anticancer effects respect to free curcumin in breast cancer cells	[171]
SLNs + CUR, d-α-Tocopheryl polyethylene glycol 1000 succinate-stabilized curcumin (TPGS) + CUR	Superior curcumin plasma levels in mice. Superior anticancer effects respect to free curcumin in Hodgkin lymphoma cells and in Hodgkin’s lymphoma xenograft models	[172]
SLNs + tristearin + PEGylated + CUR	Superior bioavailability, absorption and long-term stability after oral administration in the rats	[174]
SLNs + NaCas + NaCas-Lac + CUR	Superior stability at pH acid and antioxidant activity with respect to free curcumin	[175]
SLNs + glyceryl monostearate + poloxamer 188 + CUR	Superior stability, solubility, cellular uptake, release and anticancer effects respect to free curcumin in breast cancer cells	[176]
**Inorganic Nanoparticles**		
MNPs + CUR	Superior cellular uptake in vitro. Superior bioavailability in vivo. Superior in vitro and in vivo therapeutic efficacy respect to free curcumin in pancreatic cancer cells and in pancreatic cancer xenografts model	[180]
Folic-acid-tagged aminated-starch-/ZnO-coated iron oxide nanoparticles + CUR	Significant controlled release of curcumin and reduced hepatic and breast cancer cells viability in vitro. Cellular uptake increase in vitro	[181]
PSMNPs + CUR	Aqueous colloidal stability, biocompatibility, high loading affinity for curcumin and better curcumin release in acidic conditions. Superior cellular uptake and anticancer effects respect to free curcumin in breast cancer cells	[182]
MNP@PEG + CUR	Higher drug release in acidic conditions, biocompatibility and low cytotoxicity at physiological pH	[183]
Silica + CUR	Good stability in aqueous medium, sustained drug release and greater anticancer properties in cervical cancer cells compared to normal fibroblasts	[184]
HA-CUR@AuNPs	Good aqueous solubility, superior cellular uptake and anticancer effects respect to free curcumin in cervical, glioma and colon cancer cells	[185]
**Liposomes**		
Liposome + CUR	Improved curcumin aqueous solubility and bioavailability in tumor-bearing mice	[192]
DMPC + CUR	Superior anticancer effects respect to free curcumin in prostate cancer cells	[193]
2-hydroxypropyl-γ-cyclodextrin/ liposome + CUR	Superior in vitro and in vivo anticancer effects respect to free curcumin in osteosarcoma cancer cells	[194]
Liposome + doxorubicin + CUR	Superior anticancer effects respect to those loaded with doxorubicin alone in colon cancer cells	[195]
Liposome + CUR	Superior anticancer effects respect to free curcumin in endometrial cancer cells	[196]
Liposome + CUR + BLED-PDT therapy	Enhancement of BLED-PDT therapy effect by curcumin liposome in lung cancer cells	[197]
**Phytosomes**		
Curcuminoids + lecithin (Meriva ^®^)	Improved absorption and clinical efficacy respect to unformulated curcuminoid mixtures	[198]
Soluplus^®^ [polyvinyl caprolactam-polyvinyl acetate polyethylene glycol graft copolymer] + CPC	Improved flowability, dissolution rate and oral bioavailability in rats	[203]

**Micelles**		
MePEG-b-PCL + CUR	Improved water solubility	[206]
MePEG-b-PCL + CUR	Improved biological half-life with respect to the free curcumin in rat models	[207]
Tween-80 micelles + CUR	Improved drug plasma concentration with respect to free curcumin in volunteers	[208]
Micellar formulation + CUR (Sol-CUR)	Superior uptake, transepithelial transport, distribution and bioavailability in colon cancer cell model	[209]
Micelles + CUR	Enhancement of aqueous solubility, stability, dissolution and permeability of curcumin formulated in micelles compared to free drug	[210]
PSBMA + CUR	Greater stability, cellular uptake and tumor cytotoxicity compared to free curcumin	[211]
MPEG-P [CL-co-PDO] + CUR	Superior encapsulation efficiency, prolonged drug release profile and antitumor effects respect to free curcumin in prostate cancer cells	[212]
Pluronic F-127 + Gelucire^®^ 44/14 micelles + CUR	Controlled curcumin release, superior oral bioavailability in vivo and in vitro antitumor effects in lung cancer cells respect to free curcumin	[213]
CUR-MPP-TPGS-MMs	Small size, high drug-loading and sustained release. Improved intestinal absorption and oral bioavailabil-ity in rats	[214]
**Curcumin/β-Cyclodextrin (β-CD) and Solid Dispersions (SDs)**		
β-CD + CUR	Superior sunlight stability and solubility respect to pure colourant	[218]
Liquid-type β-CD + CUR	Improved solubility in water and bioavailability	[220,221]
Solid type β-CD + CUR	Improved storage stability and biovailability	[222]
CUR-CD-CS	Superior solubility, cellular absorption and antitumor effects compared to free curcumin in skin cancer cells	[223]
β-CD + CUR	Improved uptake and therapeutic efficacy in prostate cancer cells	[154]
β-CD + CUR	Improved delivery and therapeutic efficacy compared to free curcumin in both in vitro lung carcinoma cell lines and in vivo mouse hepatoma xenograft models	[224]
β-CD + CUR	Superior anticancer effects respect to free curcumin in cervical cancer cells	[225]
FA-CUR-NPs	Improved drug release rate, cellular uptake efficiency and in vitro and in vivo antitumor activity respect to free curcumin in cervical cancer cells	[226]
Hydroxypropyl-β-CD+ CUR+piperine	Improved solubility of the curcumin–piperine system, its permeability through biological membranes, antioxidant and antimicrobial activities and enzymatic inhibition	[227]
SDs (with Gelucire^®^50/13-Aerosil) + CUR	Improved stability, water solubility, dissolution rate, bioavailability and anti-inflammatory activity in rats	[229]
Solutol^®^ HS15 SDs + CUR	Improved solubility and bioavailability compared to free curcumin	[231]
SDs (with cellulose acetate and mannitol) + CUR	Improved water solubility and oral bioavailability compared to free curcumin	[232]
SDs + CUR	Superior water solubility and gastrointestinal absorption in rats	[233]
SDs (with Poloxamer 407) + CUR	Improved water dispersibility and cytotoxic effects against breast, lung, cervical and hepatocellular cancer cells	[234]
**Curcumin Conjugates Formulations**		
Piperine + CUR	Curcumin enhanced serum levels, reduced elimination half-life and clearance, increased bioavailability in rats and humans	[236]
Piperine + CUR	Curcumin increased bioavailability in epileptic rats	[237]
Piperine + CUR	Curcumin enhanced intestinal absorption and bioavailability in rats	[238]
BCM-95CG (Biocurcumax^®^): piperine + lecithin + CUR	Curcumin improved bioavailability and pharmacokinetic profile respect to free drug in healthy subjects	[239]

## Data Availability

Not applicable.

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
