# Peer review of "Antitumoral Activities of Curcumin and Recent Advances to ImProve Its Oral Bioavailability"

_biomedicines, 2021, doi:10.3390/biomedicines9101476_

Round 1

Reviewer 1 Report

Numerous papers and reviews are dedicated to curcumin effects on different cells and curcumin is now regarded as a promising therapeutic compound in many chronic diseases. In this comprehensive and well-written review the authors collected and critically analyzed antitumor activities of curcumin.

Manuscript could be recommended for publication after minor revision.

  1. In table 1 it should be possible to combine some data (for example first and second l. could be combine in one), then table will be shortened and more easy to read.
  2. 10, l. 241. “bladder cancer” The authors probably mean bladder cancer cells, cells should be added. The same for l. 251, “several cancers”, it should clear indicated the cell types.
  3. 11, l. 277. LPA is not mentioned in the abbreviations.
  4. 11, l. 289 - 291. The sentence “It was recently demonstrated… , should be modified.
  5. 11, l. 308. “prevent invasion by preventing” , better by inhibition.
  6. Please, correct the fonts in several places (P.21, l. 533, P. 23, l. 614, and some other places)  

Author Response

Comments and Suggestions for Authors

Numerous papers and reviews are dedicated to curcumin effects on different cells and curcumin is now regarded as a promising therapeutic compound in many chronic diseases. In this comprehensive and well-written review the authors collected and critically analyzed antitumor activities of curcumin.

Manuscript could be recommended for publication after minor revision.

In table 1 it should be possible to combine some data (for example first and second l. could be combine in one), then table will be shortened and more easy to read.

-Thank you for your suggestion. The correction has been made.

10, l. 241. “bladder cancer” The authors probably mean bladder cancer cells, cells should be added. The same for l. 251, “several cancers”, it should clear indicated the cell types.

-Thank you for your suggestion. The corrections have been made.

11, l. 277. LPA is not mentioned in the abbreviations.

-Thank you for your suggestion. The correction has been made.

11, l. 289 - 291. The sentence “It was recently demonstrated… , should be modified.

-Thank you for your suggestion. The sentence has been modified.

11, l. 308. “prevent invasion by preventing” , better by inhibition.

-Thank you for your suggestion. The correction has been made.

Please, correct the fonts in several places (P.21, l. 533, P. 23, l. 614, and some other places) 

Thank you for your suggestion. The corrections have been made.

Reviewer 2 Report

In this manuscript entitled: “Antitumoral activities of curcumin and recent advances to improve its oral bioavailability Μarta Claudia Nocito et al. present an exquisite review of the recent advances on curcumin’s interference with a number of signaling pathways related to cell cycle regulation, apoptosis and migration in cancer cell lines. The authors also provide thorough data on a number of curcumin-based derivatives/analogues as well as different drug delivery approaches that have been developed so as to overcome limiting issues such as low absorption, rapid metabolism and poor bioavailability of curcumin.

Author Response

English language and style

( ) Extensive editing of English language and style required
( ) Moderate English changes required
(x) English language and style are fine/minor spell check required
( ) I don't feel qualified to judge about the English language and style

 Comments and Suggestions for Authors

In this manuscript entitled: “Antitumoral activities of curcumin and recent advances to improve its oral bioavailability” Μarta Claudia Nocito et al. present an exquisite review of the recent advances on curcumin’s interference with a number of signaling pathways related to cell cycle regulation, apoptosis and migration in cancer cell lines. The authors also provide thorough data on a number of curcumin-based derivatives/analogues as well as different drug delivery approaches that have been developed so as to overcome limiting issues such as low absorption, rapid metabolism and poor bioavailability of curcumin.

Thank you for your suggestion. The English revision has been made.